# EGOTWIN: DREAMING BODY AND VIEW IN FIRST PERSON

**Jingqiao Xiu**[1]    **Fangzhou Hong**[2]    **Yicong Li**[1*]    **Mengze Li**[3]
**Wentao Wang**[4]    **Sirui Han**[3*]    **Liang Pan**[4*]    **Ziwei Liu**[2]

[1]National University of Singapore    [2]Nanyang Technological University
[3]Hong Kong University of Science and Technology    [4]Shanghai AI Laboratory

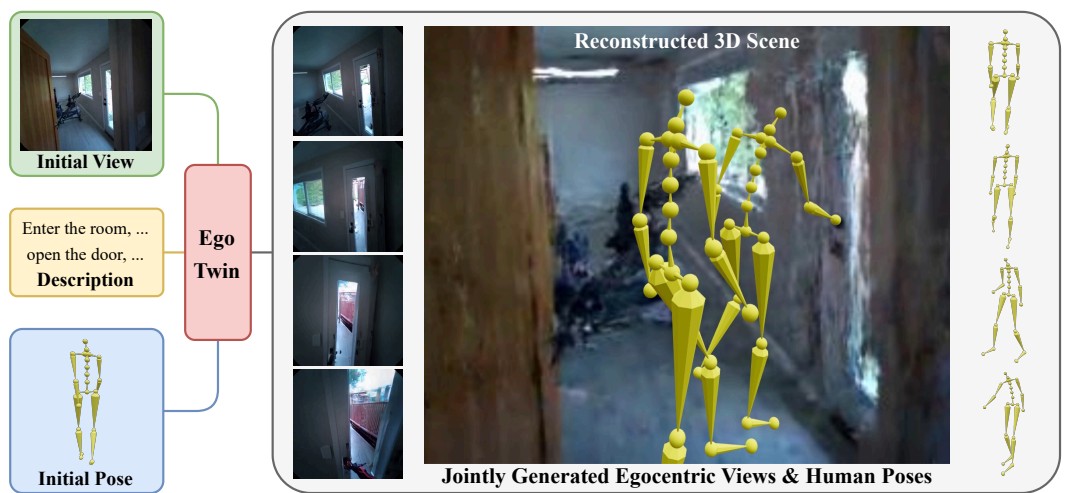

Figure 1: We propose EgoTwin, a diffusion-based framework that jointly generates egocentric video and human motion in a viewpoint consistent and causally coherent manner. Generated videos can be lifted into 3D scenes using camera poses derived from human motion via 3D Gaussian Splatting.

## ABSTRACT

While exocentric video synthesis has achieved great progress, egocentric video generation remains largely underexplored, which requires modeling first-person view content along with camera motion patterns induced by the wearer's body movements. To bridge this gap, we introduce a novel task of joint egocentric video and human motion generation, characterized by two key challenges: 1) Viewpoint Alignment: the camera trajectory in the generated video must accurately align with the head trajectory derived from human motion; 2) Causal Interplay: the synthesized human motion must causally align with the observed visual dynamics across adjacent video frames. To address these challenges, we propose EgoTwin, a joint video-motion generation framework built on the diffusion transformer architecture. Specifically, EgoTwin introduces a head-centric motion representation that anchors the human motion to the head joint and incorporates a cybernetics-inspired interaction mechanism that explicitly captures the causal interplay between video and motion within attention operations. For comprehensive evaluation, we curate a large-scale real-world dataset of synchronized text-video-motion triplets and design novel metrics to assess video-motion consistency. Extensive experiments demonstrate the effectiveness of the EgoTwin framework. Qualitative results are available on our project page: https://egotwin.pages.dev/.

## 1 INTRODUCTION

Recent advances in deep generative models have delivered remarkable progress in exocentric (third-person) video generation (Blattmann et al., 2023a; Brooks et al., 2024; Yang et al., 2025), capable of producing photorealistic and temporally consistent videos conditioned on various modalities, with broad implications for multimodal learning (Xiu et al., 2024; 2025a;b; 2026). However,

---

*Corresponding authors.

egocentric (first-person) video synthesis remains largely underexplored, despite its increasing importance for wearable computing (Fiannaca et al., 2014), augmented reality (Ashtari et al., 2020), and embodied agents (Nair et al., 2022). In contrast to exocentric setups, where the camera is static or externally controlled (Wang et al., 2024b; He et al., 2025), egocentric video captures the perspective of a moving individual, with the footage inherently entangled with the camera wearer's motion. In particular, head movements influence the camera's position and orientation, while full-body actions affect the wearer's body pose and the surrounding scene, collectively shaping the egocentric recording. Therefore, to model body-driven dynamics in egocentric views, we argue that the visual stream must be generated in lockstep with the motion stream that drives it.

In this paper, we introduce a novel task of joint video-motion generation that explicitly models egocentric video together with the motion of the camera wearer. As illustrated in Figure 1, given a static human pose and an initial scene observation, our goal is to generate synchronized sequences of egocentric video and human motion, guided by the textual description. This task introduces two fundamental challenges beyond prior works: **(1) Viewpoint Alignment.** Throughout the sequence, the camera trajectory captured in egocentric video must precisely align with the head trajectory derived from human motion. This requirement naturally stems from the fact that the camera is rigidly mounted on the wearer's head (Engel et al., 2023; Apple Inc., 2023), causing head movement and camera motion to be tightly coupled. However, existing exocentric video generation methods typically employ a unidirectional viewpoint-conditioning strategy that synthesizes video based on predefined camera poses (Wang et al., 2024b; He et al., 2025). Such approaches are unsuitable for our setting, as the camera poses in egocentric video are not externally provided but are inherently determined by the wearer's head motion. As a result, the camera poses must be generated concurrently with the human motion, necessitating a bidirectional interaction to ensure viewpoint alignment. **(2) Causal Interplay.** At each time step, the current visual frame provides spatial context that shapes human motion synthesis; conversely, the newly generated motion influences subsequent video frames. Take the "opening door" scenario in Figure 1 as an example: egocentric observation informs the wearer of the door's location, which guides the wearer's action. In turn, the performed action can alter the body pose (e.g., reaching for the doorknob), the camera pose (e.g., orienting toward the door), and the surrounding scene (e.g., the door gradually opening). These changes must be accurately reflected in subsequent video frames, thereby affecting future motion generation. This recursive dependency forms a closed observation–action loop between video and motion, highlighting the necessity of modeling their causal interplay over time.

To address these challenges, we propose EgoTwin, a joint video-motion generation framework that generates egocentric videos with body-induced camera motion patterns while capturing the causal interplay between visual observations and human actions. Specifically, EgoTwin adopts a diffusion transformer backbone (Peebles & Xie, 2023; Esser et al., 2024), with three modality-specific branches for text, video, and motion, respectively. To model the joint distribution, EgoTwin employs asynchronous diffusion in video and motion branches, which allows each modality to evolve on its timestep while maintaining cross-modal interaction. To facilitate accurate viewpoint alignment, we depart from the commonly used root-centric motion representation (Guo et al., 2022a), which obscures head pose within full-body motion and thus fails to expose the egocentric perspective to the video branch. Instead, we introduce a head-centric motion representation that anchors the human motion to the head joint, allowing for direct alignment between the camera viewpoint of the generated video and the head pose in the synthesized motion. To faithfully capture the causal interplay, we draw inspiration from the observation-action feedback loop in cybernetic systems (Agrawal et al., 2016; Pathak et al., 2017), where observations shape actions and actions alter future observations. We implement this principle through a structured interaction mechanism: each video token attends to preceding motion tokens, capturing how current observations arise from past actions, while each motion token attends to current and upcoming video tokens, enabling the inference of actions based on perceived scene transitions. This bidirectional design allows motion-driven video synthesis and video-informed motion synthesis to evolve in synchrony.

To foster research in this field, we curate a large-scale dataset of real-world egocentric videos with human pose annotations from Nymeria (Ma et al., 2024). For evaluation, we extend beyond the individual quality of video and motion, and propose video-motion consistency metrics that quantify their cross-modal alignment. Extensive experiments demonstrate the effectiveness of EgoTwin.

In summary, our contributions are fourfold:

- To the best of our knowledge, we are the first to explore the joint generation of egocentric video and human motion in a viewpoint consistent and causally coherent manner.
- We identify the limitations of conventional root-centric motion representations in egocentric contexts and reformulate a head-centric approach that facilitates video-motion alignment.
- We design a triple-branch diffusion transformer featuring a video-motion interaction mechanism, supported by an efficient three-stage training paradigm and versatile sampling strategies.
- We propose video-motion consistency metrics and build a benchmark for evaluating joint video-motion generation, where our EgoTwin demonstrates strong performance.

## 2 RELATED WORK

**Video Generation.** Video generation has witnessed significant advancements with the emergence of video diffusion models (Ho et al., 2020; Karras et al., 2022; Ho et al., 2022). A central research focus has been on text-to-video (T2V) generation and image-to-video (I2V) generation, where models synthesize coherent video sequences from textual prompts or static images. Early approaches (Blattmann et al., 2023a;b) augment UNet-based text-to-image (T2I) models (Rombach et al., 2022) with temporal modeling layers to efficiently transform them to video generation models. Recent works (Brooks et al., 2024; Yang et al., 2025) adopt transformer-based architectures (Peebles & Xie, 2023), achieving improved temporal consistency and generation quality. To incorporate camera control, representative methods (Wang et al., 2024b; He et al., 2025) inject camera parameters (e.g., extrinsic matrices or Plücker embeddings (Sitzmann et al., 2021)) into pretrained video diffusion models (Xing et al., 2024; Guo et al., 2024b). These approaches rely on known camera trajectories and encode them as input conditions. In contrast, our work considers a fundamentally different setting where the camera trajectory is not available beforehand, yet the generated video must maintain consistency with other synthesized content that is strongly correlated to the underlying camera motion. This key distinction renders existing methods inapplicable, necessitating a framework for controllable video generation that operates without predefined camera guidance.

**Motion Generation.** Generating realistic and diverse human motions from text remains a longstanding challenge in computer vision and graphics, offering intuitive control of motion synthesis through natural language. Early works (Guo et al., 2020; Petrovich et al., 2021; Guo et al., 2022a; Petrovich et al., 2022) employ temporal VAEs (Kingma et al., 2014) to capture temporal dependencies and learn probabilistic mappings between language and motion. Recent advances have introduced powerful generative modeling techniques to this field, including diffusion models (Tevet et al., 2023; Zhang et al., 2024; Chen et al., 2023), autoregressive models (Guo et al., 2022b; Zhang et al., 2023; Jiang et al., 2023), and generative masked models (Guo et al., 2024a; Pinyoanuntapong et al., 2024; Meng et al., 2025). To comply with these frameworks, motion data is represented in different forms. Diffusion-based methods typically operate on continuous vectors, either in the latent space of a VAE or directly from raw motion sequences. Autoregressive models, by contrast, often discretize motion into tokens using vector quantization techniques such as VQ-VAE (Van Den Oord et al., 2017) or RVQ-VAE (Lee et al., 2022). Generative masked models are flexible in this regard, accommodating both discrete and continuous representations depending on the loss function and model architecture. Furthermore, several researchers (Hassan et al., 2021; Wang et al., 2021; Huang et al., 2023; Zhao et al., 2023) have investigated human motion generation within 3D scenes represented as RGB point clouds. Others combine the above two tasks by simultaneously incorporating textual and scene information (Wang et al., 2022; Cen et al., 2024; Wang et al., 2024a; Yi et al., 2024). Our work differs from this line of research in how scene information is provided: instead of granting full scene access during motion synthesis, we observe the scene only once from the initial human pose and rely on a generative model to hallucinate scene observations as the human moves. In egocentric vision, studies (Li et al., 2023; Yi et al., 2025; Hong et al., 2025) focus on estimating the human motion from the egocentric video, while we explore this close relationship through joint generative modeling.

**Multimodal Generation.** Recent advances have expanded generative models from unimodal to multimodal generation. Specifically for diffusion models, Ruan et al. (2023) introduces the first multimodal diffusion framework for synchronized audio-video generation. Other studies (Xu et al., 2023; Bao et al., 2023) design unified models capable of jointly generating text and images. In the domain of human motion, Li et al. (2024) pioneers the simultaneous generation of motion and frame-level language descriptions that explain the generated motions. Despite these developments, the joint modeling of human motion and its corresponding egocentric views remains largely unexplored. To the best of our knowledge, we take the first step in this direction, uncovering their tight coupling.

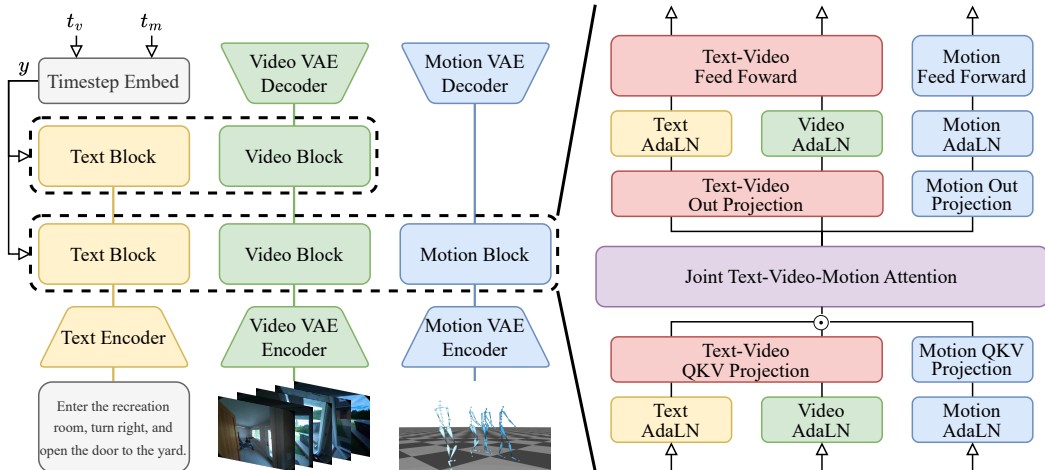

Figure 2: EgoTwin features a triple-branch architecture (**left**), where the motion branch spans only the lower half of the layers used by text and video branches. Each branch has its own tokenizer and transformer blocks (**right**), with shared weights across branches indicated by matching colors.

## 3 METHODOLOGY

**Problem Definition.** Given an initial human pose $P^0 \in \mathbb{R}^{J \times 3}$ in a scene, an egocentric observation $I^0 \in \mathbb{R}^{H \times W \times 3}$ from that pose, and a textual description of intended human actions in the scene, our goal is to generate two synchronized sequences: (1) a human pose sequence $P^{1:N_m} \in \mathbb{R}^{N_m \times J \times 3}$ and (2) an egocentric view sequence $I^{1:N_v} \in \mathbb{R}^{N_v \times H \times W \times 3}$ spanning the same duration. Here, $J$ is the number of human joints, $H$ and $W$ are the image height and width, $N_m$ and $N_v$ are the number of frames in the pose and view sequences, respectively. This forms a closed-loop generation paradigm where video and motion mutually and continuously influence each other over time.

**Framework Overview.** An overview of EgoTwin is shown in Figure 2. Text, video, and motion inputs are first encoded using a text encoder, a video VAE encoder, and a motion VAE encoder, respectively. These embeddings are then processed through the corresponding branches of a diffusion transformer. Finally, the video and motion outputs are decoded by respective VAE decoders.

### 3.1 MODALITY TOKENIZATION

For the text and video modalities, we adopt T5-XXL (Raffel et al., 2020) as the text tokenizer and encoder, and a 3D causal VAE (Yang et al., 2025) as the video tokenizer. Specifically, the input text is first tokenized and adjusted to a fixed length $L_t$ via truncation or padding, then encoded into text embeddings $c \in \mathbb{R}^{L_t \times D_t}$. The video frames are temporally and spatially compressed into latent representations $z_v \in \mathbb{R}^{\left(\frac{N_v}{4}+1\right) \times \frac{H}{8} \times \frac{W}{8} \times C_v}$ with a compression ratio of $4 \times 8 \times 8$ and $C_v$ latent channels, which are subsequently patchified and unfolded into video embeddings $X_v \in \mathbb{R}^{L_v \times D_v}$ of sequence length $L_v$. $D_t$ and $D_v$ denote the embedding dimension of text and video, respectively.

**Motion Representation.** Unlike the uniform representation for text and video, motion representation exhibits a great degree of diversity. Currently, the most widely adopted format in human motion generation is the overparameterized canonical pose representation (Guo et al., 2022a), which has become the default standard for popular datasets, including KIT-ML (Plappert et al., 2016) and HumanML3D (Guo et al., 2022a). Formally, the human pose at each frame is defined as a tuple of $(\dot{r}^a, \dot{r}^{xz}, r^y, j^p, j^v, j^r, c^f)$, comprising seven groups of features: root angular velocity along Y-axis $\dot{r}^a$, root linear velocities on XZ-plane $\dot{r}^{xz}$, root height $\dot{r}^y$, local joint positions $j^p \in \mathbb{R}^{3(J-1)}$ and velocities $j^v \in \mathbb{R}^{3(J-1)}$ in root space, joint rotations $j^r \in \mathbb{R}^{6(J-1)}$ in local space, and binary foot-ground contacts $c^f \in \mathbb{R}^4$. Motions are retargeted to a default human skeletal template and initially rotated to face the positive Z-axis.

However, the above root-centric representation is not suitable for our task, as the critical information for alignment with the egocentric video, such as the pose of the head joint, is deeply buried in a multi-step kinematic calculation. Mathematically, recovering the head joint pose requires integrating root velocities to obtain the root pose, then applying forward kinematics (FK) to propagate transformations through the kinematic chain to the head joint. Intuitively, this computation is too complex to be precisely modeled by neural networks, as experimentally substantiated in Section C.

To address this issue, we propose a head-centric motion representation that explicitly exposes egocentric information. Specifically, we define the representation as a tuple $(h^r, \dot{h}^r, h^p, \dot{h}^p, j^p, j^v, j^r)$, where $h^r \in \mathbb{R}^6$ and $\dot{h}^r \in \mathbb{R}^6$ are the absolute and relative rotation of the head joint, $h^p \in \mathbb{R}^3$ and $\dot{h}^p \in \mathbb{R}^3$ are the absolute and relative position of the head joint. The terms $j^p$ and $j^v$ are now expressed in head space, while $j^r$ retains its original meaning. Additionally, we normalize the initial head pose to zero translation and identity rotation, and set all first-order kinematic features to zero in the initial frame. Our representation naturally resonates with egocentric video in at least two novel ways: 1) It offers more accurate access to the head trajectory, which closely correlates with camera movement; 2) It more clearly informs the egocentric video how the body is observed egocentrically.

**Motion Tokenization.** Inspired by the Causal 3D CNN (Yu et al., 2024), we build the motion VAE using 1D causal convolutions, where all padding is applied at the beginning of the convolutional axis. The encoder and decoder are symmetrically structured, each comprising two stages of $2\times$ downsampling or upsampling, interleaved with ResNet blocks (He et al., 2016). The motion VAE is trained using a combination of reconstruction loss $\mathcal{L}_{rec}$ and Kullback–Leibler (KL) divergence regularization $\mathcal{L}_{\mathrm{KL}}$ weighted by $\lambda_{\mathrm{KL}}$. To ensure that loss contributions are balanced across different groups regardless of their dimensions, we compute the VAE loss $\mathcal{L}_{\mathrm{VAE}}$ separately for the 3D head $(h^p, \dot{h}^p)$, 6D head $(h^r, \dot{h}^r)$, 3D joint $(j^p, j^v)$, and 6D joint $(j^r)$ components. The final loss averages these four items:

$$\mathcal{L}_{\mathrm{VAE}} = \frac{1}{4} \sum_c \left( \mathcal{L}_{rec}^{(c)} + \lambda_{\mathrm{KL}} \mathcal{L}_{\mathrm{KL}}^{(c)} \right), \text{where } c \in \{\text{head}_{3D}, \text{head}_{6D}, \text{joint}_{3D}, \text{joint}_{6D}\}. \tag{1}$$

Using the trained VAE, motion representations are tokenized into latents $Z_m \in \mathbb{R}^{\left(\frac{N_m}{4}+1\right) \times C_m}$ with a $4\times$ downsampling rate and $C_m$ channels, and subsequently transformed into motion embeddings $X_m \in \mathbb{R}^{L_m \times D_m}$, with $L_m$ as the sequence length and $D_m$ as the embedding dimension.

## 3.2 DIFFUSION TRANSFORMER

Our diffusion transformer extends MM-DiT (Esser et al., 2024), initially designed for text-to-image generation, to support text, video, and motion modalities. As illustrated in Figure 2, each branch consists of a sequence of MLPs and applies adaptive layer normalization (AdaLN) in conjunction with a gating mechanism (Peebles & Xie, 2023) to incorporate timestep information. The text and video branches are initialized from CogVideoX (Yang et al., 2025), with shared weights except for the AdaLNs. The motion branch corresponds to only the lower half of the layers in other branches, as essential visual cues for video-motion interaction, such as camera pose and scene structure, are primarily captured in the early layers of the video diffusion backbone. In contrast, the higher layers specialize in appearance details, which are less relevant to motion. To further improve efficiency, the motion branch employs reduced channel dimensions, consistent with the lower representational complexity of motion relative to video. The embedding sequences from different modalities are projected to a common dimensionality $D$ and concatenated for joint attention operations (Vaswani et al., 2017). This triple-branch architecture allows each modality to work in its own representational space while still attending to and interacting with the others.

**Interaction Mechanism.** The original MM-DiT framework includes only text and image modalities, where cross-modal consistency is enforced only at the global level, i.e., matching the entire image with the entire text suffices. However, our task demands fine-grained temporal synchronization between video and motion: each video frame must be temporally aligned with the corresponding motion frame. Although we incorporate sinusoidal positional encodings (Vaswani et al., 2017) for both video and motion tokens, along with 3D rotary position embeddings (RoPE) (Su et al., 2024) for video tokens to provide absolute and relative position information, these mechanisms primarily capture intra-modal structure. Consequently, the inter-modal correspondence at each time step remains implicit to the diffusion transformer, which may lead to globally consistent outputs that nevertheless lack frame-wise synchronization.

To address this challenge, we explicitly encode the causal interplay between video and motion by introducing a structured joint attention mask to the diffusion transformer. Given that human motion is typically captured at a higher temporal resolution than egocentric video, we set the number of motion tokens to be twice the number of video tokens (i.e., $N_m = 2N_v$), without loss of generality.

Formally, we follow the notations in Cybernetics (Agrawal et al., 2016; Pathak et al., 2017) to rewrite $I^i$ as the observation $O^i$, and $(P^{2i+1}, P^{2i+2})$ as the (chunked) action $A^i$, where $i \in [0, N_v - 1]$. According to the principles of forward dynamics: $\{O^i, A^i\} \rightarrow O^{i+1}$ and inverse dynamics: $\{O^i, O^{i+1}\} \rightarrow A^i$, video tokens corresponding to $O^i$ can attend to motion tokens that correspond to $A^{i-1}$, capturing how $O^i$ arise from $A^{i-1}$, while motion tokens corresponding to $A^i$ can attend to video tokens that correspond to both $O^i$ and $O^{i+1}$, enabling the inference of $A^i$ based on scene transitions from $O^i$ to $O^{i+1}$. A special case is given to $P^0$, which is allowed bilateral attention with $I^0$. As demonstrated in Figure 3, apart from the aforementioned relationship, the remaining attention between video and motion is blocked, while all intra-modal attention, as well as inter-modal attention related to text, are preserved.

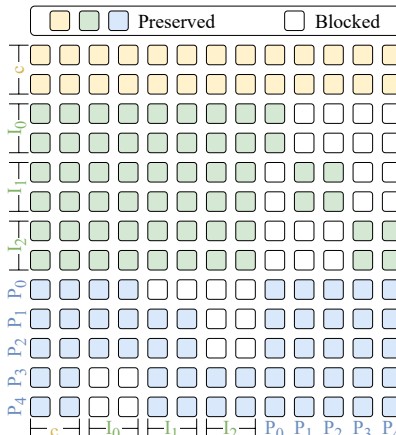

Figure 3: Interaction mechanism.

**Asynchronous Diffusion.** We independently sample two timesteps, $t_v$ and $t_m$, between $0$ and $T$ (maximum timestep), and add Gaussian noises $\epsilon_v$ and $\epsilon_m$ associated with these timesteps to the latents $z_v$ and $z_m$, respectively. Each timestep is first encoded via a sinusoidal embedding, and an MLP then processes two concatenated embeddings to produce a unified timestep embedding $y$, which serves as input to the AdaLN layers. Our model consists of a video denoiser $\epsilon_\theta^v(z_v^{t_v}, z_m^{t_m}, c, t_v, t_m)$ and a motion denoiser $\epsilon_\theta^m(z_m^{t_m}, z_v^{t_v}, c, t_m, t_v)$, which are jointly optimized to simultaneously predict the noises added to the video and motion latents using the following objective:

$$\mathcal{L}_{\text{DiT}} = \mathbb{E}_{\epsilon_v, \epsilon_m, c, t_v, t_m} \left[ \left\| \epsilon_v - \epsilon_\theta^v(z_v^{t_v}, z_m^{t_m}, c, t_v, t_m) \right\|_2^2 + \left\| \epsilon_m - \epsilon_\theta^m(z_m^{t_m}, z_v^{t_v}, c, t_m, t_v) \right\|_2^2 \right]. \quad (2)$$

### 3.3 TRAINING AND SAMPLING

**Training Paradigm.** Our training schema comprises three stages: 1) *Motion VAE Training*, as described in Equation (1). 2) *Text-to-Motion Pretraining*. Since the motion branch lacks pretrained weights for initialization, we pretrain it on the text-to-motion task using only text and motion embeddings as input, while keeping the text branch frozen. Following classifier-free guidance (CFG) (Ho & Salimans, 2022), we randomly discard the text embeddings with a probability of 10% to model unconditional motion generation. By omitting the much longer video embeddings, we can leverage greater parallelism, which accelerates the training process. Critically, freezing the text branch not only preserves the pretrained text-to-video weights but also facilitates the integration of motion embeddings into the pretrained text-video embedding space. 3) *Joint Text-Video-Motion Training*, as formulated in Equation (2). Video embeddings are incorporated in this final stage, and the model learns the joint distribution of video and motion conditioned on text. Again, text embeddings are randomly dropped with a probability of 10% to model the unconditional video-motion generation.

**Sampling Strategy.** Benefiting from the joint distribution modeling, our framework supports not only joint video-motion generation conditioned on text (T2VM), but also unimodal generation, including video generation conditioned on text and motion (TM2V), and motion generation conditioned on text and video (TV2M). The CFG for TM2V sampling is defined as follows:

$$\hat{\epsilon}_\theta^v(z_v^t, z_m^0, c, t, 0) = \epsilon_\theta^v(z_v^t, z_m^T, \phi, t, T) + w_t \left( \epsilon_\theta^v(z_v^t, z_m^T, c, t, T) - \epsilon_\theta^v(z_v^t, z_m^T, \phi, t, T) \right)$$
$$+ w_m \left( \epsilon_\theta^v(z_v^t, z_m^0, c, t, 0) - \epsilon_\theta^v(z_v^t, z_m^T, c, t, T) \right). \quad (3)$$

The CFG formula for TV2M sampling can be derived by exchanging the roles of $v$ and $m$ in Equation (3). Here, $w_t$, $w_v$, and $w_m$ denote the guidance scales for text, video, and motion conditions, respectively. For T2VM sampling, taking the motion branch as an example (with the video branch being analogous), its CFG formula is expressed as:

$$\hat{\epsilon}_\theta^m(z_m^t, z_v^t, c, t, t) = \epsilon_\theta^m(z_m^t, z_v^T, \phi, t, T) + w_t \left( \epsilon_\theta^m(z_m^t, z_v^T, c, t, T) - \epsilon_\theta^m(z_m^t, z_v^T, \phi, t, T) \right)$$
$$+ w_v \left( \epsilon_\theta^m(z_m^t, z_v^t, c, t, t) - \epsilon_\theta^m(z_m^t, z_v^T, c, t, T) \right). \quad (4)$$

After sampling, latents from the video branch are unpatchified to recover their original shape and then decoded by the 3D causal VAE decoder (Yang et al., 2025) to reconstruct the video, while latents from the motion branch are passed through the motion VAE decoder to reconstruct the motion.

## 4 EXPERIMENTS

### 4.1 EVALUATION METRICS

**Video Quality.** We adopt Image Fréchet Inception Distance (**I-FID**) (Heusel et al., 2017) to evaluate the visual fidelity and realism of individual frames by measuring the distributional distance between the features of generated frames and those of real images. At the video level, we employ Fréchet Video Distance (**FVD**) (Unterthiner et al., 2018) to quantify temporal coherence and consistency across generated video sequences compared to real ones. Additionally, CLIP Similarity (**CLIP-SIM**) (Wu et al., 2021) is utilized to assess the semantic alignment and contextual relevance between generated video clips and textual prompts.

**Motion Quality.** We choose Motion Fréchet Inception Distance (**M-FID**) (Heusel et al., 2017) to assess the statistical similarity between the high-level features of generated motions and real motions. To evaluate the alignment between text and motion, we train a GRU-based text feature extractor and a GRU-based motion feature extractor, both sharing the same architecture as the evaluator in (Guo et al., 2022a). These models are optimized using a contrastive loss on GloVe (Pennington et al., 2014) text embeddings and our motion representation described in Section 3.1, ensuring that matched text-motion pairs yield geometrically close feature vectors. Within this learned feature space, the text-to-motion Retrieval Precision (**R-Prec**) is measured in terms of Top-3 retrieval accuracy. Meanwhile, the Multimodal Distance (**MM-Dist**) captures the average Euclidean distance between corresponding motion and text features.

**Video-Motion Consistency.** We propose to evaluate the consistency between generated egocentric videos and human motions from two aspects: 1) *View Consistency*: We first estimate the frame-wise camera poses of the generated egocentric videos using DROID-SLAM (Teed & Deng, 2021) and extract the head joint poses from the generated human motions. Then, we align both trajectories at the first frame and apply Procrustes Analysis to determine the optimal scale factor that aligns the estimated camera trajectory with the extracted head trajectory. Finally, we compute the Translation Error (**TransErr**) as the average Euclidean distance between the corresponding camera and head positions, and the Rotation Error (**RotErr**) as the average angular difference between the corresponding camera and head orientations, using the same formulas as He et al. (2025). 2) *Hand Consistency*: We detect the presence of the left and right hands, equipped with the motion capture device, in the generated egocentric videos. For the generated human motions, we compute the hand visibility from the perspective of a virtual camera mounted on the corresponding head joint with known intrinsics. Based on the presence and visibility analysis, we define the Hand F-Score (**HandScore**) as the average F-Score of left and right hands, where a *True Positive* means the hand is present in the video and visible from the head in motion, a *False Positive* means the hand is present in the video but invisible from the head in motion, and a *False Negative* means the hand is absent in the video but visible from the head in motion.

### 4.2 EXPERIMENTAL SETUP

**Dataset.** To overcome the limitations of using synthetic or small-scale real-world datasets for evaluation, we train and evaluate our model on Nymeria (Ma et al., 2024), a large-scale, real-device dataset that captures diverse people engaged in a wide range of daily activities across various indoor and outdoor locations. The dataset provides paired text-video-motion data, including egocentric videos recorded with Project Aria glasses (Engel et al., 2023), full-body motions captured using the Xsens inertial motion capture system (Paulich et al., 2018), and motion narrations written by human annotators. All data are segmented into 5-second clips, yielding approximately 170K samples after filtering, which are split into training, validation, and test sets for the joint training stage. We ensure that both the individuals and environments in the test split remain unseen during joint training.

**Baseline.** Since no prior methods are capable of addressing our task, we propose a simple yet effective baseline, VidMLD, that retains the architecture of EgoTwin while removing all dedicated designs introduced in Section 3.1 and Section 3.2. In other words, VidMLD combines the state-of-the-art video diffusion model CogVideoX (Yang et al., 2025) and the latent-space motion diffusion model MLD (Chen et al., 2023), both of which excel in unimodal generation, and connects them through the multimodal diffusion architecture MM-DiT (Esser et al., 2024) to enable joint generation. We adopt the same three-stage training recipe described in Section 3.3, and employ the original classifier-free guidance (Ho & Salimans, 2022) for sampling.

Table 1: Quantitative results of joint video and motion generation, evaluated by metrics covering video quality, motion quality, and video-motion consistency.

| Method | Video Quality | | | Motion Quality | | | Video-Motion Consistency | | |
|---|---|---|---|---|---|---|---|---|---|
| | I-FID ↓ | FVD ↓ | CLIP-SIM ↑ | M-FID ↓ | R-Prec ↑ | MM-Dist ↓ | TransErr ↓ | RotErr ↓ | HandScore ↑ |
| VidMLD | 157.86 | 1547.28 | 25.58 | 45.09 | 0.47 | 19.12 | 1.28 | 1.53 | 0.36 |
| EgoTwin | 98.17 | 1033.52 | 27.34 | 41.80 | 0.62 | 15.05 | 0.67 | 0.46 | 0.81 |

**Prompt:** Enter the recreation room, turn right, and open the door to the yard.

**Prompt:** Turn left to walk into the kitchen, then turn towards the living area.

Figure 4: Qualitative results of joint video and motion generation, based on a textual prompt and initial frames of both video and motion.

**Implementation Details.** In our experiments, videos are undistorted and resized to a resolution of $H = W = 480$, with each segment containing $N_v + 1 = 41$ frames at 8 FPS. The motion data adopts the Xsens skeleton with $J = 23$ joints and consists of $N_m + 1 = 81$ frames per segment at 16 FPS. The video and motion latents have $C_v = 16$ and $C_m = 64$ channels, respectively. The embedding lengths for text, video, and motion are $L_t = 226$, $L_v = 9900$, and $L_m = 21$, with corresponding dimensions $D_t = D_v = D = 3072$, and $D_m = 768$. The hyperparameter $\lambda_{KL}$ in Equation (1) is set to 1e-4. CFG scales are set to $w_t = 6$ for text and $w_v = w_m = 4$ for video and motion. The text and video branches have 42 layers, totaling approximately 5B parameters, with most shared across both branches. The motion branch comprises 21 layers, corresponding to the lower halves of the other two branches, and contains roughly 300M parameters.

## 4.3 MAIN RESULTS

**Quantitative Results.** As shown in Table 1, EgoTwin significantly outperforms the baseline method across all evaluation metrics, with especially pronounced improvements in video-motion consistency scores. The brute-force joint training of the video and motion generation models leads to poor alignment between the two modalities, resulting in notably lower video-motion consistency performance. In contrast, EgoTwin effectively captures the intrinsic correlation between the two modalities, achieving not only excellent cross-modal consistency but also enhanced single-modal generation quality through the mutually beneficial interaction between video and motion modalities.

**Qualitative Results.** We visualize several examples generated by EgoTwin in Figure 4. These samples illustrate that the video and motion streams not only adhere to the textual descriptions for single-modal generation but also evolve in strict cross-modal synchrony, particularly in terms of camera viewpoint and head pose, as well as in scene content and human action. We encourage readers to visit our project page (https://egotwin.pages.dev/) for richer generation examples.

## 4.4 ABLATION STUDIES

We present the extensive ablation studies in Table 2, where each row corresponds to a specific ablation setting. All variants exhibit a consistent performance decline across all metrics compared to our full model (listed at the bottom), confirming the effectiveness of each design. First, we replace our Motion Reformulation with the standard representation (Guo et al., 2022a) commonly used in human motion generation research ("w/o MR"). The resulting performance drop highlights the importance of our reformulation in exposing egocentric motion cues to the video, which fundamentally facilitates the alignment between egocentric video and human motion. Next, we remove the Interaction Mechanism from the joint attention operations and instead apply full attention without masking

Table 2: Ablation results on three designs: Motion Reformulation (MR), Interaction Mechanism (IM), and Asynchronous Diffusion (AD).

| Variant | Video Quality | | | Motion Quality | | | Video-Motion Consistency | | |
|---|---|---|---|---|---|---|---|---|---|
| | I-FID ↓ | FVD ↓ | CLIP-SIM ↑ | M-FID ↓ | R-Prec ↑ | MM-Dist ↓ | TransErr ↓ | RotErr ↓ | HandScore ↑ |
| w/o MR | 134.27 | 1356.81 | 26.36 | 43.65 | 0.56 | 17.31 | 0.96 | 1.22 | 0.44 |
| w/o IM | 117.54 | 1237.58 | 27.10 | 44.01 | 0.59 | 15.87 | 0.85 | 0.89 | 0.57 |
| w/o AD | 109.73 | 1124.19 | 26.91 | 42.58 | 0.53 | 16.48 | 0.74 | 0.62 | 0.73 |
| EgoTwin | 98.17 | 1033.52 | 27.34 | 41.80 | 0.62 | 15.05 | 0.67 | 0.46 | 0.81 |

Table 3: Comparisons between joint video–motion modeling and separate video/motion modeling.

| Method | Video Quality | | | Motion Quality | | |
|---|---|---|---|---|---|---|
| | I-FID ↓ | FVD ↓ | CLIP-SIM ↑ | M-FID ↓ | R-Prec ↑ | MM-Dist ↓ |
| CogVideoX | 182.97 | 1793.79 | 24.90 | – | – | – |
| CameraCtrl | 120.48 | 1263.90 | 27.01 | – | – | – |
| MLD | – | – | – | 47.25 | 0.39 | 21.47 |
| EgoTwin | 98.17 | 1033.52 | 27.34 | 41.80 | 0.62 | 15.05 |

("w/o IM"). The observed degradation underscores its critical role in capturing causal relationships between video and motion, as well as ensuring fine-grained temporal synchronization. Finally, we substitute the Asynchronous Diffusion with a synchronous counterpart for video and motion latents, and accordingly simplify the sampling algorithm to vanilla CFG ("w/o AD"). The performance decline validates its value for modeling comprehensive and diverse dependencies between video and motion modalities, and enabling precise textual control over the joint generation process.

## 4.5 IN-DEPTH ANALYSIS

To demonstrate the advantages of joint modeling, we compare our joint generation (Text-to-Video-Motion) with separate generation approaches (Text-to-Video and Text-to-Motion), with results reported in Table 3. We also implement and compare against a camera-controlled video generation method based on CogVideoX. The substantial performance gains confirm that joint modeling significantly enhances the generation quality of each modality. This is corroborated by Table 1, which shows that the stronger joint generation capability of EgoTwin yields notable improvements in both video and motion metrics compared to VidMLD, which exhibits weaker joint modeling.

Beyond quantitative evaluation, we also provide a qualitative analysis using the door-opening example in Figure 1. Without modeling the state of the door at each timestep and its temporal evolution, the generated human motion often appears unrealistic, as it lacks awareness of the door. On the other hand, without modeling the underlying human motion that drives the egocentric video, the resulting viewpoint shifts and scene dynamics tend to be physically implausible. Moreover, relying solely on camera poses as external control signals fails to capture the underlying human motion in egocentric video, underscoring the necessity for internal motion modeling in egocentric video generation.

## 4.6 CROSS-DATASET EVALUATION

Given the scarcity of high-quality datasets synchronizing full-body human motion with egocentric video, our evaluation in Table 1 centers on the Nymeria dataset. Specifically, we establish a rigorous evaluation protocol by strictly partitioning the data to ensure that human subjects and interaction scenes in the test set are unseen during the joint training stage. To further assess the generalization capabilities of the video-motion correspondence captured by our model, we conduct a systematic cross-dataset evaluation on Ego-Exo4D (Grauman et al., 2024). However, a substantial discrepancy exists in skeletal representations: Ego-Exo4D annotations adopt the sparse 17-keypoint COCO format (Lin et al., 2014), which structurally differs from the dense 23-joint Xsens skeleton required by our model, rendering direct inference infeasible.

Table 4: Cross-dataset results of joint video and motion generation, evaluated by metrics covering video quality, motion quality, and video-motion consistency.

| Method | Video Quality | | | Motion Quality | | | Video-Motion Consistency | | |
|---|---|---|---|---|---|---|---|---|---|
| | I-FID ↓ | FVD ↓ | CLIP-SIM ↑ | M-FID ↓ | R-Prec ↑ | MM-Dist ↓ | TransErr ↓ | RotErr ↓ | HandScore ↑ |
| VidMLD | 173.95 | 1738.15 | 24.27 | 52.18 | 0.37 | 23.09 | 1.46 | 1.78 | 0.32 |
| EgoTwin | 115.82 | 1242.60 | 26.13 | 48.34 | 0.50 | 20.16 | 0.70 | 0.51 | 0.79 |

**Prompt:** Open and close the kitchen cabinet.

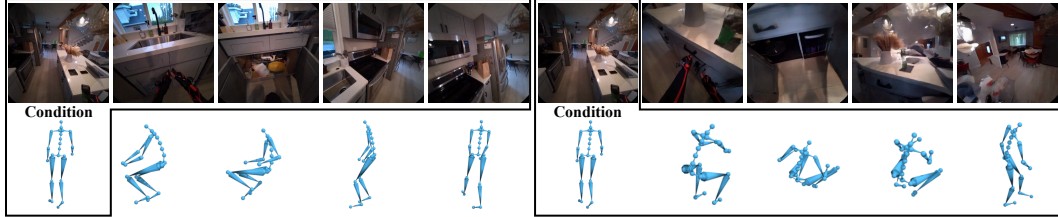

Figure 5: Results of conditional generation. **Left:** motion generation conditioned on text and video; **Right:** video generation conditioned on text and motion.

To address this skeletal incompatibility, we design an optimization-based retargeting pipeline. Specifically, we first regress the sparse COCO keypoints to the SMPL parametric model (Loper et al., 2015), leveraging its inherent statistical shape and pose priors to reconstruct missing degrees of freedom and enforce anatomical plausibility. Subsequently, we retarget the resolved SMPL kinematics to the Xsens rig via precise joint remapping and spline-based interpolation. Finally, we employ forward kinematics to propagate local joint angles into global joint positions and rotations. As evidenced in Table 4, EgoTwin not only significantly outperforms the baseline but also maintains superior video-motion consistency metrics. This validates that the video-motion correspondence learned by EgoTwin is robust and generalizes effectively to external datasets.

### 4.7 BROADER APPLICATIONS

**Conditional Generation.** Our joint distribution enables conditional sampling of one modality given another, using the CFG algorithm described in Equation (3). As shown in Figure 5, we can generate human motion conditioned on text and egocentric video (left), as well as generate egocentric video conditioned on text and human motion (right). Interestingly, textual descriptions are often ambiguous (e.g., they may refer to cabinets on the left or right side of the scene in Figure 5), the ability to additionally condition on either motion or video provides greater control over the generation process, which further substantiates the strong coupling between video and motion in our model.

**Scene Reconstruction.** With jointly generated video and motion, we can effortlessly extract camera poses from human motion and directly integrate both modalities into a 3D Gaussian Splatting (Kerbl et al., 2023) pipeline. As illustrated in Figure 1, we reconstruct the 3D scene from the generated video and seamlessly position the synthesized human into it by aligning head poses with camera trajectories. The realistic spatial interactions exhibited, such as the feet on the ground and the right hand near the door handle, demonstrate strong spatiotemporal coherence between the generated egocentric videos and human motion.

## 5 CONCLUSION

We propose EgoTwin, a diffusion-based framework that jointly generates egocentric video and human motion in a viewpoint consistent and causally coherent manner. Our method introduces a head-centric motion representation and a cybernetics-inspired interaction mechanism, supported by an efficient three-stage training paradigm and versatile sampling strategies. To evaluate our approach, we establish a comprehensive benchmark that includes a large-scale dataset of text-video-motion triplets and novel video–motion consistency metrics. Experiments demonstrate that EgoTwin delivers promising results. We hope our work encourages further exploration of joint generative modeling for egocentric video and human motion, and lays a solid foundation for future research in this area.

ACKNOWLEDGMENTS

This work is funded in part by the HKUST Start-up Fund (R9911), Theme-based Research Scheme grant (T45-205/21-N), the InnoHK initiative of the Innovation and Technology Commission of the Hong Kong Special Administrative Region Government, and the research funding under HKUST-DXM AI for Finance Joint Laboratory (DXM25EG01).

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

# APPENDIX

## A   DETAILS OF MOTION VAE

**VAE Encoder.** The encoder begins with a 1D causal convolution (kernel size 3) that projects the 288-channel motion input to 512 channels, followed by four DownBlocks that progressively expand the channel dimension to 2048. The first two DownBlocks each downsample the temporal dimension by a factor of 2 using 1D causal convolutions with a kernel size of 4 and a stride of 2. Each DownBlock comprises four ResNet layers, with each layer including two 1D causal convolutions (kernel size 3), GroupNorm, SiLU activation, and optional shortcut connections for channel alignment. A MidBlock comprising two 2048-channel ResNet layers further processes the features. The encoder ends with a 1D causal convolution (kernel size 3) that outputs 128 channels, representing the mean and log-variance of a 64-dimensional motion latent space.

**VAE Decoder.** The decoder mirrors the encoder in reverse. It starts with a 1D causal convolution (kernel size 3) that maps the 64-dimensional motion latent vector to 2048 channels, followed by a MidBlock with two 2048-channel ResNet layers. Four UpBlocks then progressively reduce the channel dimension to 512. The first two UpBlocks each upsample the temporal dimension by a factor of 2 using 1D nearest neighbor interpolation and 1D causal convolution (kernel size 3). Each UpBlock includes five ResNet layers, structurally identical to those in the encoder. The decoder ends with a 1D causal convolution (kernel size 3) that reconstructs the original 288-channel motion representation. Both the encoder and decoder contain approximately 200M parameters.

## B   DETAILS OF CONSISTENCY METRICS

Since the generated motion sequence has twice the temporal resolution of the video sequence, we downsample it by selecting every other frame to match the length of the video sequence, $N_v$, before evaluating the consistency metrics described below.

**View Consistency.** We denote the canonicalized egocentric camera trajectory and head joint trajectory (i.e., with each frame expressed relative to the first frame) as sequences of rotation matrices and translation vectors: $[R_v \mid T_v] \in \mathbb{R}^{N_v \times 3 \times 4}$ and $[R_m \mid T_m] \in \mathbb{R}^{N_v \times 3 \times 4}$, respectively. Let $s$ denote the optimal scale factor for scene scale alignment. The Translation Error (TransErr) is calculated as the scale-invariant $L_2$ distance between the translation vector sequences $T_v$ and $T_m$:

$$\text{TransErr} = \|sT_v - T_m\|_2^2. \tag{5}$$

The Rotation Error (RotErr) is computed by comparing the rotation matrix sequences $R_v$ and $R_m$:

$$\text{RotErr} = \arccos\left(\frac{\text{tr}(R_v R_m^T) - 1}{2}\right), \tag{6}$$

where $\text{tr}(\cdot)$ denotes the trace of a matrix. Since each sequence is normalized with respect to the first frame, errors are computed as the average frame-wise error over the remaining frames.

**Hand Consistency.** Because the camera wearer's hands are covered by motion capture gloves rather than bare skin, state-of-the-art hand landmark detection tools, such as MediaPipe (Lugaresi et al., 2019) and HaMeR (Pavlakos et al., 2024), fail to detect the wearer's hands in video frames. To overcome this limitation, we formulate the problem as two independent binary classification tasks: one for detecting the presence of the left hand and the other for the right hand in each frame. Our hand classifier is built upon a pretrained Vision Transformer (ViT) (Dosovitskiy et al., 2021) backbone and is trained using supervision signals derived from human poses and camera intrinsics. To mitigate the issue of class imbalance between positive and negative samples, we adopt Focal Loss (Lin et al., 2017) as our classification objective, which is defined as:

$$\mathcal{L}_{\text{CLS}} = \sum_{c=1}^{2} -\alpha_t^c \left(1 - p_t^c\right)^\gamma \log\left(p_t^c\right), \tag{7}$$

where $\gamma$ is the focusing parameter, $c \in \{0, 1\}$ indicates the hand side, $p_t^c$ is the predicted probability for the target class $t$, and $\alpha_t^c$ is a weighting factor associated with the target class $t$. Given the

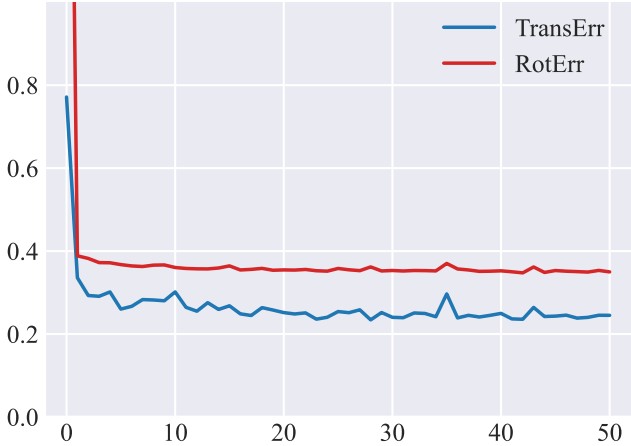

Figure 6: Head pose regression errors over epochs.

predicted probability $p = \mathrm{sigmoid}(x)$ obtained from the output logits of the model $x \in \mathbb{R}^2$, and the ground-truth labels $y \in \{0, 1\}^2$, the values of $p_t^c$ and $\alpha_t^c$ are calculated as:

$$p_t^c = \begin{cases} p^c & \text{if } y^c = 1 \\ 1 - p^c & \text{if } y^c = 0 \end{cases}, \quad \alpha_t^c = \begin{cases} \alpha^c & \text{if } y^c = 1 \\ 1 - \alpha^c & \text{if } y^c = 0 \end{cases}. \tag{8}$$

In our experiments, we set $\alpha^c = 0.80$ for the left hand and $\alpha^c = 0.75$ for the right hand, and use a focusing parameter of $\gamma = 2$.

## C  DETAILS OF REGRESSION EXPERIMENT

To validate our insights in Section 3.1, we train a GRU-based regression model that takes the root-centric motion representation sequences as input, supervised by an MSE loss against the ground-truth head pose sequences. As shown in Figure 6, both translation and rotation errors (TransErr and RotErr, see Section 4.1 for details) plateau at high levels, due to insufficient explicit cues for accurately modeling head pose.

## D  ADDITIONAL QUALITATIVE RESULTS

**Joint Generation.** Figure 7 presents additional examples generated by EgoTwin. Together with those in Figure 4, these results demonstrate the model's capacity to synthesize a variety of motion types (e.g., walking, opening, closing, grabbing) across diverse environments (e.g., bedrooms, kitchens, living areas, outdoor yards) and involving a wide range of objects of interaction (e.g., cabinets, clothing, doors, drawers). Video demonstrations can be found on our project page: https://egotwin.pages.dev/.

**Conditional Generation.** Recall that in Figure 5, we demonstrate that when the textual description is ambiguous concerning the referent objects involved in the interaction, conditional sampling strategies (i.e., TV2M or TM2V) allow for fine-grained control over the generation process. Another salient source of linguistic ambiguity arises from human interaction behaviors. For instance, while the prompt in Figure 8 clearly specifies the target object of interaction (i.e., the pillow located on the right side of the sofa in Figure 8), it leaves the manner of the human action unconstrained (e.g., whether the individual uses the left hand or right hand). In such cases, incorporating additional conditioning signals can help disambiguate these subtleties.

**Failure Cases.** The failure cases primarily stem from the physical implausibility of the generated videos. For instance, in the first example of Figure 7, the cup displays abnormal deformation. This issue reflects a broader challenge in current video generation models and is difficult to address through joint modeling with human motion, as the motion signal typically captures only the global displacement of the object. Nevertheless, we generally observe strong correspondence between the generated video and motion.

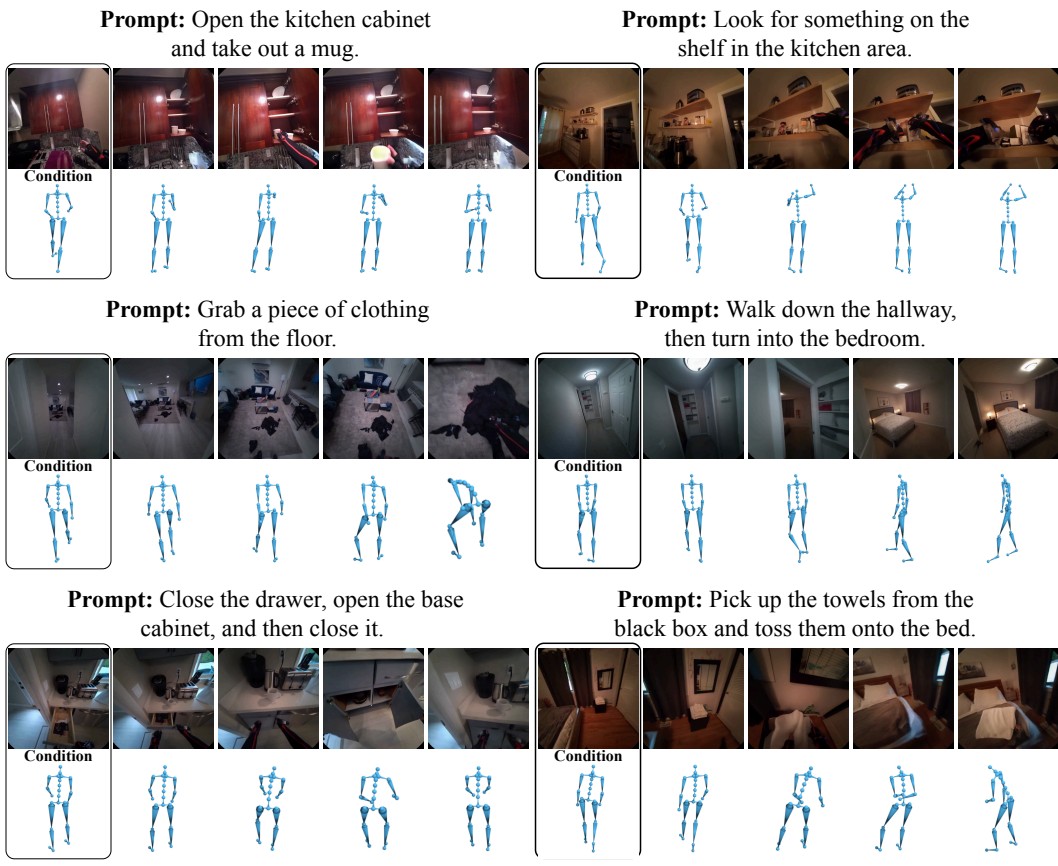

Figure 7: Additional qualitative results of joint generation, based on a textual prompt and initial frames of both video and motion.

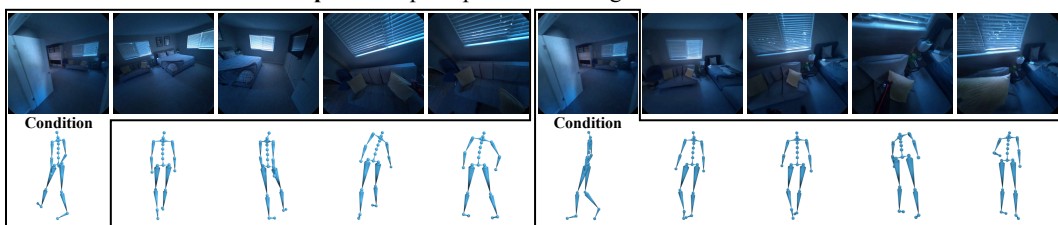

Figure 8: Additional qualitative results of conditional generation. **Left:** motion generation conditioned on text and video; **Right:** video generation conditioned on text and motion.

## E    LIMITATIONS AND FUTURE WORKS

One major limitation of EgoTwin is its focus on full-body motion representation without incorporating hand joints. This stems from the absence of hand motion capture data in our dataset, which restricts our model's ability to generate hand movements. While our model achieves reasonable consistency between generated video and motion, the synthesized videos may still exhibit artifacts that violate physical laws or lack 3D coherence, posing challenges for downstream applications.

Our future work could involve finetuning on datasets that include hand motion to enable the generation of coordinated body and hand motions alongside egocentric videos. Additionally, integrating 3D priors and physical constraints may enhance the realism of generated videos, thereby improving the fidelity of the corresponding motion synthesis. Other avenues for improvement include increasing video resolution and extending the temporal context to support longer sequences.

