# OpenReview forum: "EgoTwin: Dreaming Body and View in First Person"
_ICLR.cc/2026/Conference — ICLR 2026 Poster_

### Official Review · Reviewer_ZiMJ · 2025-10-30

**Soundness:** 4
**Presentation:** 4
**Contribution:** 3
**Rating:** 8
**Confidence:** 4

**Summary:**

This paper introduces EgoTwin, a diffusion framework for the novel task of jointly generating egocentric video and the wearer's full-body motion. The method tackles the key challenges of Viewpoint Alignment (matching camera to head) and Causal Interplay (the feedback loop between seeing and acting) . The architecture is a triple-branch (text, video, motion) Diffusion Transformer featuring two main contributions: a head-centric motion representation for direct alignment and a cybernetics-inspired attention mask to model causality. Using the Nymeria dataset and new consistency metrics, the paper demonstrates strong performance.

**Strengths:**

- The paper introduces a novel and clearly defined problem of *joint egocentric video and human motion generation*, with strong motivation and potential applications in embodied AI and AR.
- The head-centric motion representation is a well-motivated and technically sound reformulation that effectively exposes egocentric cues for video-motion alignment.
- The model design—a triple-branch diffusion transformer with asynchronous diffusion and causal attention—is elegant and verified through systematic ablations.
- Comprehensive evaluation, including novel *video-motion consistency metrics*, supports the technical claims. The qualitative results are convincing.

**Weaknesses:**

- The paper relies exclusively on the Nymeria dataset. While uniquely suited, this makes it difficult to assess model robustness. A brief justification for not using other large-scale egocentric datasets (e.g., EgoExo4D) would be helpful.
- All data are segmented into 5-second clips. It's unclear if this is a dataset constraint (narration length) or a computational limit of the diffusion model. A discussion on this limitation and the path toward modeling longer action sequences would be beneficial.
- The degree to which the text prompts are "free-form" is not analyzed. A description of the text diversity in the Nymeria dataset would be helpful to understand the model's linguistic generalization
- The main paper's baseline comparison (Table 1) is limited to one method. The more informative comparison against strong unimodal baselines (video-only and motion-only) is in the appendix (Table 3). Maybe the authors could move Table 3 to the main paper to better highlight the benefits of joint modeling.

**Questions:**

This is a very strong and interesting paper. My main questions for clarification are the weaknesses listed above.

---

> ### Author Response · Authors · 2025-11-24
>
> We sincerely thank you for acknowledging the strong motivation and potential applications of our introduced problem, the technical soundness of our method design, and the comprehensive evaluation.
>
> **Weakness 1:**
>
> Our decision to focus on Nymeria was driven by two primary factors:
> - **Data Quality**: We prioritized the Nymeria dataset because its motion data is derived from professional motion capture systems, providing high-fidelity, low-noise ground truth. In contrast, the motion annotations in Ego-Exo4D are generated via algorithmic estimation, which introduces unavoidable noise and artifacts.
> - **Skeleton Incompatibility**: A significant technical barrier to using Ego-Exo4D is the structural disparity. Our model is based on the 23-joint Xsens skeleton used by Nymeria, whereas Ego-Exo4D adopts 17 COCO keypoints, making direct training and evaluation incompatible.
>
> To conduct cross-dataset evaluation, we utilize the Ego-Exo4D dataset by retargeting the annotated COCO keypoints to the Xsens skeleton. We provide the details of this evaluation in the newly added Section 4.6 of our revised manuscript.
>
> **Weakness 2:**
>
> The 5-second clip duration is a constraint dictated by the Nymeria dataset, as the associated text narrations are annotated at 5-second intervals. Nevertheless, to synthesize longer sequences, we can investigate autoregressive training paradigms such as Self-Forcing [1] to adapt EgoTwin for long-video generation.
>
> **Weakness 3:**
>
> The original Nymeria paper provides a comprehensive analysis of the linguistic annotations. Specifically, Section 3.4 details their free-form annotation process, and Figure 7 presents a word cloud visualizing the frequency and variety of terms used. This confirms that the text descriptions cover a diverse range of actions and scenes.
>
> **Weakness 4:**
>
> We agree that the comparison against strong unimodal baselines is vital for demonstrating the advantages of our joint modeling. In our revision, we have moved Table 3 to the main paper.
>
> [1] Xun Huang, et al. Self Forcing: Bridging the Train-Test Gap in Autoregressive Video Diffusion, NeurIPS 2025.

---

### Official Review · Reviewer_Dv46 · 2025-10-30

**Soundness:** 2
**Presentation:** 2
**Contribution:** 3
**Rating:** 4
**Confidence:** 2

**Summary:**

This paper proposed EgoTwin, a newly defined task that joins the generations of egocentric (first-person) video and human motion, addressing two core challenges: Viewpoint Alignment (ensuring camera trajectories in generated videos match head trajectories from human motion) and Causal Interplay (modeling influences between visual results and human actions). Formally, the authors proposed a multi-modal generation model to tackle this task, including a motion branch and a video generation branch. For the motion generation, a head-centric motion representation is proposed, while the Joint Text-Video-Motion Attention is used to unify both modalities. Experiments show the effectiveness of the proposed method.

**Strengths:**

1. To the best of my knowledge, EgoTwin is the first work to explicitly model joint egocentric video and human motion generation.
2. Several useful and reasonable techniques are proposed in this paper to handle this joint generation. The head-centric motion representation directly solves the limitation of root-centric representations. The attention mechanism is adjusted to improve the multi-modal learning and address causal interplay.

**Weaknesses:**

1. The main concern is the unclear motivation. The paper does not sufficiently justify why a complex triple-branch diffusion generation (with specialized motion generation/video interaction modules) is required, especially given recent advances in scaling video foundation models (e.g., Genie3) that can generate high-fidelity, interactive egocentric videos via implicit scene and motion modeling. EgoTwin heavily depends on explicit motion representation and cross-modal modules, and well-labeled data (Nymeria), which may not be necessary if foundation models can implicitly address these issues with less annotation and better generalization. While Nymeria is large (170K samples in 5s), the diversity and generalization are still questionable.

2. The illustration of the Interaction Mechanism is chaotic and confusing. What is the meaning of "human motion is typically captured at a higher temporal resolution than egocentric video", and why set $N_m = 2N_v$? The presentation of $O$ and $A$ are not well defined. What is the relation among action $A$, observation $O$, pose $P$, and view sequence $I$? Figure 3 is also very confusing without a proper illustration.

**Questions:**

The authors should clarify the motivation and generalizability of the proposed approach, and providing more details about the Interaction Mechanism would be helpful.

---

> ### Author Response · Authors · 2025-11-24
>
> We sincerely thank you for recognizing EgoTwin as the first work to explicitly model joint egocentric video and human motion generation, and our technical contributions. We will address your concerns point by point below.
>
> **Weakness 1:**
>
> We want to clarify that EgoTwin addresses several key distinctions not met by state-of-the-art video foundation models (e.g., Genie3):
> - **Action Space Expressiveness**: Genie3’s action space is constrained to simple directional movements (e.g., navigation). In contrast, EgoTwin operates on a far more complex and expressive action space representing human motion.
> - **Dynamic Scene Interaction**: Genie3’s demonstrations for embodied research primarily showcase agent navigation without dynamic scene interaction. In contrast, EgoTwin can generate high-fidelity dynamic interactions with objects in the scene.
> - **Instruction Guided Autonomy**: Genie3 simulates future states based on a sequence of low-level actions; consequently, achieving a goal requires manually decomposing it into specific navigation steps. Conversely, EgoTwin operates as an autonomous agent driven by high-level language instructions, generating both the human motion towards the goal and the egocentric video along the way.
>
> In summary, by providing explicit 3D skeletal structures beyond video generation, EgoTwin extends the capabilities of video models in the above aspects and facilitates downstream research in embodied AI, such as imitation learning.
>
> Regarding the generalizability, we evaluate this in two ways. First, our evaluation on Nymeria employs strict data partitioning, where the test set contains only novel subjects and scenes held out from the joint training stage. As shown in Table 1, EgoTwin effectively captures video-motion alignment in these unseen scenarios. Second, we conduct cross-dataset evaluation on Ego-Exo4D by retargeting the annotated COCO keypoints to the Xsens skeleton. We provide the details of this evaluation in the newly added Section 4.6 of our revised manuscript.
>
> **Weakness 2:**
>
> The setting of $N_m = 2N_v$ models the inherent discrepancy between the high temporal frequency of human motion and the lower frame rates of video. It reflects both **hardware standards** (e.g., 120 fps motion capture vs. 30 fps video) and **physical reality**, where multiple intermediate motion states accumulate to produce a single observable change in the subsequent video frame. Without loss of generality, we model this frequency ratio as 2.
>
> The variables $O, A, P, I$ represent a notation mapping defined in Lines 270-272. We rewrite the video frame $I_i$ as the Observation $O_i$. Simultaneously, we define the Action $A_i$ as the aggregate of human poses $(P_{2i+1}, P_{2i+2})$ that occur between observation $O_i$ and the next observation $O_{i+1}$.
>
> Figure 3 illustrates the specific attention mask designed to enforce causal coherence. A video token attends only to the preceding motion that causes it. Conversely, a motion token attends to both the current and next observations. This restricted pattern is supported by the principles of forward dynamics (predicting the next observation) and inverse dynamics (inferring the action taken).

---

### Official Review · Reviewer_9aiA · 2025-10-31

**Soundness:** 3
**Presentation:** 4
**Contribution:** 4
**Rating:** 8
**Confidence:** 3

**Summary:**

This paper introduces EgoTwin, a generative approach that synthesizes ego-centric video and human poses jointly based on text instructions. It presents a novel motion representation for aligning viewpoints from a head-centric perspective. To model the relationships among text, motion, and video, the paper introduces an advanced interaction mechanism. Experiments on video quality, motion quality, and video-motion consistency underline the effectiveness of the proposed method.

**Strengths:**

1. The paper is well-written and easy to follow, with the main idea clearly articulated. The implementation details are sufficient for reproduction.
2. The design of head-centric motion tokenization is novel.
3. The interaction mechanism is reasonable; it employs local temporal attention, which not only improves accuracy but also reduces computational load.

**Weaknesses:**

1. While head-centric motion tokenization may enhance head-centric evaluations, the quality of the whole body is not assessed. Can a full-body evaluation be included?
2. A related paper is not cited: https://egoallo.github.io.

**Questions:**

1. What does "c" represent in Figure 3?
2. I don't fully understand why two different time steps for video and motion generation are necessary. It appears that a single time step would suffice for joint motion and video generation. What considerations influenced this design choice?
3. Regarding unimodal sampling in Equation 3, I understand that "T" is set to zero on the left-hand side due to elimination. However, why is the variable "T" on the right-hand side? Should it be sampled over a grid of "t" and "T"?
4. Similarly, in Equation 4, why does the left-hand side use both "t" and "t," while the right-hand side uses "t" and "T"?

---

> ### Author Response · Authors · 2025-11-24
>
> We sincerely thank you for recognizing the novelty and presentation of our paper. We will address your concerns point by point below.
>
> **Weakness 1:**
>
> We would like to clarify that our motion evaluation is performed on the full body. The three motion quality metrics used in our evaluation assess the generated full-body motion against the ground-truth full-body motion.
>
> **Weakness 2:**
>
> In the revision, we have discussed this paper and the associated line of work in Section 2.
>
> **Question 1:**
>
> The variable $c$ in Figure 3 refers to the text embedding, as defined in Line 197.
>
> **Question 2:**
>
> Our design choice to use two different time steps was influenced by two main considerations:
> - **Modeling Diverse Dependencies**: Using different time steps for the video and motion denoisers allows the model to capture more comprehensive dependencies between the video and motion modalities. The advantages of this approach are empirically validated in our ablation study (Table 2).
> - **Enabling Conditional Sampling**: Crucially, this design enables flexible conditional sampling, where the conditioning modality is clean while the generated modality starts from noise. This requires the noise levels of the two modalities to differ during training, which is achieved via asynchronous diffusion. Since the two denoisers are exposed to various combinations of relative noise, the "noisier" denoiser learns to reference the "cleaner" one, which enables conditional sampling.
>
> **Question 3:**
>
> Equation 3 describes the Classifier-Free Guidance (CFG) formula for sampling a video at time step $t$, conditioned on both text and motion. Therefore, the video component is at time step $t$ in all terms, and the conditional motion on the left-hand side is $0$ (i.e., clean). We apply the CFG twice. First, sampling is conditioned only on text; in this case, the motion modality is treated as noise, so its time step is $T$. The second application introduces motion as an additional condition. This explains why the motion component in the second line of Equation 3 involves both step $0$ (conditional) and step $T$ (unconditional).
>
> **Question 4:**
>
> Equation 4 describes the CFG formula for sampling motion at time step $t$ during joint sampling. Here, the motion component is at time step $t$ in all terms. The video component on the left-hand side is also at $t$, as it represents the current state of the joint sampling. Similar to Equation 3, we apply the CFG twice. The first term on the right-hand side conditions only on text, treating the video modality as noise (time step $T$). The second term introduces the video as an additional condition, which is why the video component in the second line of Equation 4 involves both $t$ (conditional) and $T$ (unconditional).

---

### Official Review · Reviewer_LqoE · 2025-11-03

**Soundness:** 3
**Presentation:** 4
**Contribution:** 3
**Rating:** 8
**Confidence:** 4

**Summary:**

The paper introduces EgoTwin, a diffusion-based generative framework for jointly producing egocentric videos and human motion that are both viewpoint consistent (ensuring alignment between camera and head movement) and causally coherent (visual observations and human actions influence each other over time). To train and evaluate the model, the authors leverage the Nymeria dataset of synchronized text–video–motion triplets and introduce video–motion consistency metrics to assess alignment between generated video and motion.

**Strengths:**

- The proposed approach, built on a triple-branch diffusion transformer, effectively models text, video, and motion within a unified framework. The cybernetics-inspired interaction mechanism is interesting, which captures the bidirectional dependencies between visual and motion streams.

- The paper is well-written, clearly structured, and easy to follow, with strong motivation and coherent presentation of technical details.

- The evaluation is extensive and thorough, covering diverse quantitative and qualitative metrics that  demonstrate the model’s performance.

**Weaknesses:**

- The approach relies exclusively on the Nymeria dataset for training and evaluation. While Nymeria is a large and well-curated dataset, the generalization of EgoTwin to other egocentric or synthetic environments remains untested, like EgoExo4D.

- The base model design choice for the text–video component relies on CogVideoX. While this is a strong foundation, it raises the question of whether using a more recent or higher-capacity base model could further improve video quality or multimodal alignment.

**Questions:**

N/A

---

> ### Author Response · Authors · 2025-11-24
>
> We sincerely thank you for acknowledging the EgoTwin framework and interaction mechanism, as well as commending our strong motivation, clear presentation, and extensive evaluation. We will address your concerns point by point.
>
> **Weakness 1:**
>
> The primary challenge in evaluating our model on other egocentric datasets, such as Ego-Exo4D, lies in the significant disparity in skeletal representations: Nymeria uses the 23-joint Xsens structure, whereas Ego-Exo4D uses 17 COCO keypoints. This makes immediate cross-dataset evaluation non-trivial. We evaluate the generalization of EgoTwin in two ways. First, our evaluation on Nymeria employs strict data partitioning, where the test set contains only novel subjects and scenes held out from the joint training stage. As shown in Table 1, EgoTwin effectively captures video-motion alignment in these unseen scenarios. Second, we conduct cross-dataset evaluation on Ego-Exo4D by retargeting the annotated COCO keypoints to the Xsens skeleton. We provide the details of this evaluation in the newly added Section 4.6 of our revised manuscript.
>
> **Weakness 2:**
>
> We agree that adopting more advanced base models is a valuable direction. Since EgoTwin is backbone-agnostic for DiT-style architectures (e.g., CogVideoX, Wan2.1), replacing the base model is feasible. Taking Wan2.1 as an example, in which the separate text branch in CogVideoX is replaced by a cross-attention mechanism, we can similarly adapt the architecture by introducing a motion branch parallel to the video branch. The cross-attention mechanism would then operate on the concatenation of video and motion tokens. We will explore the integration of EgoTwin with these advanced models to enhance generation quality.

---

### Author Response · Authors · 2025-12-01
**Summary of Responses**

We sincerely thank the reviewers for their constructive feedback, which has significantly strengthened our work. Below, we summarize our responses to the key concerns raised and outline the specific revisions made to the manuscript.

**1. Evaluation of Model Generalization (R-LqoE W1, R-Dv46 W1, R-ZiMJ W1)**
- Cross-dataset evaluation: To address concerns regarding dataset reliance, we introduced a cross-dataset evaluation on Ego-Exo4D. We bridged the skeletal discrepancy (17 COCO keypoints vs. 23 Xsens joints) by retargeting the COCO keypoints to the Xsens structure, enabling a direct assessment of our model on this external dataset.
- Generalization to unseen scenarios: We clarified that our Nymeria evaluation employs a strict split where the test set contains only novel subjects and scenes held out from training. As shown in Table 1, EgoTwin effectively generalizes to these unseen scenarios.
- Rationale for dataset choice: We justified our prioritization of Nymeria due to its professional, high-fidelity motion capture data, which offers a cleaner signal compared to the algorithmically estimated annotations in Ego-Exo4D.

**2. Clarification of Evaluation and Methodology (R-9aiA W1 & Q1-4, R-Dv46 W1-2)**
- Evaluation metrics: We clarified that our motion evaluation metrics assess full-body motion fidelity, rather than just head-centric movement.
- Dual time step design: We explained that using distinct time steps for video and motion generation allows the model to capture diverse dependencies and is essential for enabling conditional sampling.
- Equation details: We provided a step-by-step explanation of the Classifier-Free Guidance (CFG) formulas (Equations 3 and 4).
- Distinctions from video models: We distinguished EgoTwin from video foundation models like Genie3. While Genie3 focuses on navigation via low-level actions, EgoTwin handles a complex action space, dynamic scene interactions, and instruction-guided autonomy. The explicit 3D skeletal output (i.e., body) specifically facilitates downstream embodied AI research (e.g., imitation learning).
- Interaction mechanism: We justified the 2:1 ratio of motion to video based on hardware standards and physical reality. We also provided explanations for the notations ($O, A, P, I$) and the attention mask design in Figure 3.

**3. Discussions of Model Extensions  (R-LqoE W2, R-ZiMJ W2)**
- Base model flexibility: We emphasized that the EgoTwin framework is backbone-agnostic. While currently built on CogVideoX, we elaborated on how the framework can readily integrate more powerful DiT-style architectures (e.g., Wan2.1).
- Longer video generation: We acknowledged that the 5-second clip duration is a dataset-imposed constraint. We discussed potential directions, such as employing autoregressive methods (e.g., Self-Forcing), to extend EgoTwin for long-video generation.

**4. Revisions to the Manuscript**
- Related Work: We updated Section 2 to include a discussion of relevant literature, including "EgoAllo".
- Experiments: We added Section 4.6 to detail the cross-dataset evaluation on Ego-Exo4D. Moreover, we moved the comparison against strong unimodal baselines (video-only and motion-only) from the Appendix to the Main Paper (Table 3).

---

### Meta-Review · Area_Chair_STco · 2025-12-27

**Summary:**

Pre-rebuttal the paper was already rated as good paper by three out of four reviewers. They appreciated the novel problem, design of the approach, quality of presentation, and strong evaluation. Their identified weaknesses are relatively minor, asking for some additional citations, more ablations, datasets, and backbone evaluations. Reviewer ZiMJ also identifies a 5-second clip dataset constraint. Reviewer Dv46 was more critical, especially on the method presentation, with a concern on unclear motivation for the complex triple-branch diffusion generation, and unclear interaction mechanism.

The author-provided rebuttal did a good job in addressing the reviewer comments. Additional experiments were added and effort was put in improving the method presentation. AC believes the presentation concerns raised by Reviewer Dv46 are addressed, and AC requests the authors to more clearly incorporate their responses in the camera ready version. This paper can be accepted for ICLR 2026.

**Reviewer Concerns:**

All reviewer concerns were relatively minor. The requested additional citations and experiments have been added, a 5-sec clip dataset limitation has been acknowledged, and overall presentation has been further improved.

**Reviewer Scores:**

I believe reviewer Dv46 would have upgraded their score from 4 to at least 6. Perhaps asking for a more explicit inclusion of the author responses in the method description section.
The three other reviewers were already high, and I believe they would have stayed at 8. Certainly no need to lower their scores.

---

### Decision · Program_Chairs · 2026-01-26

Accept (Poster)